

# Biomechanical analyses of pterygotid sea scorpion chelicerae uncover predatory specialisation within eurypterids

Russell D. C. Bicknell[1,2], Yuri Simone[3], Arie van der Meijden[3], Stephen Wroe[1,2], Gregory D. Edgecombe[4] and John R. Paterson[1]

[1] Palaeoscience Research Centre, School of Environmental & Rural Science, University of New England, Armidale, NSW, Australia
[2] Function, Evolution and Anatomy Research Lab, School of Environmental and Rural Science, University of New England, Armidale, NSW, Australia
[3] CIBIO Research Centre in Biodiversity and Genetic Resources, Vila do Conde, Portugal
[4] Science Group, The Natural History Museum, London, UK

Corresponding author
Russell D. C. Bicknell,
rdcbicknell@gmail.com

## ABSTRACT

Eurypterids (sea scorpions) are extinct aquatic chelicerates. Within this group, members of Pterygotidae represent some of the largest known marine arthropods. Representatives of this family all have hypertrophied, anteriorly-directed chelicerae and are commonly considered Silurian and Devonian apex predators. Despite a long history of research interest in these appendages, pterygotids have been subject to limited biomechanical investigation. Here, we present finite element analysis (FEA) models of four different pterygotid chelicerae—those of *Acutiramus bohemicus*, *Erettopterus bilobus*, *Jaekelopterus rhenaniae*, and *Pterygotus anglicus*—informed through muscle data and finite element models (FEMs) of chelae from 16 extant scorpion taxa. We find that *Er. bilobus* and *Pt. anglicus* have comparable stress patterns to modern scorpions, suggesting a generalised diet that probably included other eurypterids and, in the Devonian species, armoured fishes, as indicated by co-occurring fauna. *Acutiramus bohemicus* is markedly different, with the stress being concentrated in the proximal free ramus and the serrated denticles. This indicates a morphology better suited for targeting softer prey. *Jaekelopterus rhenaniae* exhibits much lower stress across the entire model. This, combined with an extremely large body size, suggests that the species likely fed on larger and harder prey, including heavily armoured fishes. The range of cheliceral morphologies and stress patterns within Pterygotidae demonstrate that members of this family had variable diets, with only the most derived species likely to feed on armoured prey, such as placoderms. Indeed, increased sizes of these forms throughout the mid-Palaeozoic may represent an 'arms race' between eurypterids and armoured fishes, with Devonian pterygotids adapting to the rapid diversification of placoderms.

Subjects Computational Biology, Evolutionary Studies, Paleontology, Zoology
Keywords Euarthropoda, Finite element analysis, Predation, Eurypterids, Sea scorpions

## INTRODUCTION

Feeding toolkits of proposed fossil predators are typically explored through functional morphology, often with comparison to modern analogues. In the last two decades, there

has been a focus on modelling animals using two-dimensional (2D) and three-dimensional (3D) biomechanical analyses, including finite element analyses (FEA) (*Ross, 2005*; *Rayfield, 2007*; *van Heteren et al., 2021*; *Rowe & Snively, 2022*). This latter method has been an effective approach for modelling fossil vertebrates and, as such, applications of FEA in palaeontology have largely focused on select vertebrate groups (*Rayfield et al., 2001*; *Wroe, McHenry & Thomason, 2005*; *Wroe et al., 2007, 2018*; *Kupczik et al., 2009*; *Strait et al., 2009*; *Attard et al., 2016*). By comparison, fossil arthropods have not been examined as thoroughly with 3D FEA (*Bicknell et al., 2018a, 2021*; *Esteve et al., 2021*). Recent focus on modelling extinct arthropods has increased knowledge of the biomechanical capability of these fossil forms.

Fossil predatory arthropods are epitomised by the large, aquatic eurypterid family Pterygotidae, known from Silurian and Lower Devonian deposits (*Braddy, Poschmann & Tetlie, 2008*; *McCoy et al., 2015*; *Lamsdell & Selden, 2017*). Some forms have bodies reaching ~2.5 m in length and represent the largest known marine arthropods, living or extinct (*Braddy, Poschmann & Tetlie, 2008*; *Lamsdell & Braddy, 2010*; *Vermeij, 2016*). A key feature of pterygotids is their large, anteriorly-directed chelicerae that often show enlarged denticles (*Ciurca & Tetlie, 2007*), likely used to capture and subdue prey (*Kjellesvig-Waering, 1964*; *Waterston, 1964*; *Miller, 2007b*; *Braddy, Poschmann & Tetlie, 2008*; *Kennedy, Miller & Gibling, 2012*; *Bicknell, Smith & Poschmann, 2020*). Given this, pterygotids are presumed to have been apex predators within their respective ecosystems (*Selden, 1984*; *Plotnick & Baumiller, 1988*; *Braddy, Poschmann & Tetlie, 2008*). Despite the striking appearance of pterygotid chelicerae, to date, biomechanical investigations of these structures have been limited to 2D lever arm studies (*Selden, 1984*; *Laub, Tollerton & Berkof, 2010*). Examination of these chelicerae using 3D biomechanical analysis could therefore present a more complete understanding of their functional morphology. Importantly, modern scorpion pedipalp chelae can inform on the mechanical performance of pterygotid chelicerae, as the former are perhaps the closest functional analogue for the fossil forms. Furthermore, as scorpions are a diverse chelicerate group that are phylogenetically closer to eurypterids than other arthropod groups with chelate appendages, such as decapod crustaceans (*Shultz, 2007*; *Legg, Sutton & Edgecombe, 2013*; *Haug, 2020*), and likely have comparable cuticular properties to eurypterids, scorpions are one of the more informative groups to function as modern analogues.

Pterygotid chelicerae are occasionally preserved with sufficient anatomical fidelity to allow for detailed reconstructions. In particular, *Acutiramus bohemicus*, *Erettopterus bilobus*, *Jaekelopterus rhenaniae*, and *Pterygotus anglicus* are four exceptionally-preserved and well-documented species, making them ideal for biomechanical analysis (*Waterston, 1964*; *Chlupáč, 1994*; *Poschmann & Tetlie, 2006*; *Miller, 2007b*; *Braddy, Poschmann & Tetlie, 2008*; *Dunlop, Penney & Jekel, 2020*). Furthermore, modern scorpions provide a diverse and well-studied group for comparative purposes (*Van der Meijden, Kleinteich & Coelho, 2012*; *Lourenço, 2021*). Building on previous FEAs of scorpions (*Van der Meijden, Kleinteich & Coelho, 2012*) and analyses of extinct arthropods (*Bicknell et al., 2021*), here we present 3D finite element models (FEMs) of the chelicerae of *Ac. bohemicus*, *Er. bilobus*, *Ja. rhenaniae*, and *Pt. anglicus* and compare them to models of chelae of 16 scorpion taxa.

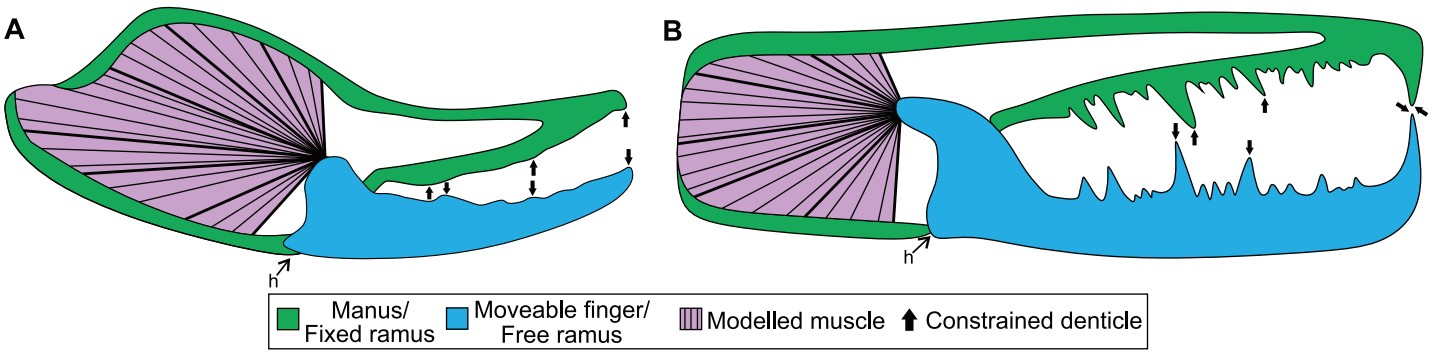

**Figure 1** **Theoretical models used for the biomechanical analyses, colour coded for analogous structures.** (A) Scorpion model. (B) Pterygotid model. Abbreviation: h, hinge. Muscle fibre organisation is used to illustrate generalised muscle directions but was not used to determine muscle force (see Methods).

## METHODS

### Extant models

Information on the muscle that adducts the moveable finger in a scorpion chela follows *Snodgrass (1952)*, *Gilai & Parnas (1970)*, and *Van der Meijden, Kleinteich & Coelho (2012)* (Fig. 1A). One primary muscle mass fills the majority of the proximal manus region and is responsible for adducting the moveable finger. There is also a secondary closing muscle in the patella (*Gilai & Parnas, 1970*). We have not considered this muscle here because: (a) scans of patellar sections needed to estimate muscle size were not made; and (b) there is no fossil evidence for this muscle (such as scars or fibres) in eurypterids. Data for calculating force of the primary closing muscle were collected from micro-computed tomography (micro-CT) scans. The scans were made with a Skyscan 1076 micro-CT scanner using a source voltage of 31 kV and a source current of 187 µA at 35 µm resolution, as detailed in *Van der Meijden, Kleinteich & Coelho (2012)*. Sixteen specimens, each representing a different scorpion genus and spanning six families, were scanned and chelae segmented with Amira/Avizo 5 (Thermo Fischer Scientific, Waltham, MA, USA). The external cuticle and internal manus content were segmented either manually or using the "Thresholding" tool in Amira. Internal manus content, excluding regions where adducting muscles were not present, was segmented as a globular structure to estimate muscle volume, surface area, and cross-sectional area. Values of volume and area were calculated with the "Surface Area Volume" module in Avizo, while the cross-sectional area was calculated by bisecting the muscle volume transversely (*i.e.*, perpendicular to the proximal-distal axis of the manus and approximately perpendicular to muscle fibre directions) in Meshlab v. 2020.12 (*Cignoni et al., 2008*). Analysed specimens were preserved in ethanol and housed in the Centro de Investigação em Biodiversidade e Recursos Genéticos, Vila do Conde, Portugal (CIBIO) collection and assigned the prefix of 'Sc'.

Maximum pinch force data were only known from the micro-CT scanned specimens of *Androctonus bicolor* and *Pandinoides cavimanus* (*Simone & Van der Meijden, 2018*). As we did not measure *in-vivo* pinch forces of the micro-CT scanned individuals of other species, pinch forces were estimated for those specimens using data available from other

individuals of the same species. As such, chela length, height, and width (*Stahnke, 1970*) were used to predict pinch force for our FEMs based on known pinch forces of other individuals of the same species (*Van der Meijden, Herrel & Summers, 2010*; *Van der Meijden, Kleinteich & Coelho, 2012*; Table S1). The data were $\log_{10}$ transformed to linearise variables with different dimensionality and used to produced species-specific linear models, and associated regression coefficients (Table S2).

The mechanical advantage of the moveable finger was calculated from the micro-CT scans following *Simone & Van der Meijden (2018)*. However, as *in-vivo* pinch forces were measured approximately two thirds of the finger length from the joint and not the distal-most point of moveable fingers, the mechanical advantage was corrected by shortening the length of the out-lever by one third. Muscle force at insertion was then calculated for scanned specimens by dividing pinch force by the specimens' mechanical advantage (Table S3). Muscle stresses used here were derived from *Van der Meijden et al. (2012)*, where the stress of *Galeodes* sp. (203 kPa) and *Rhagodes* sp. (905 kPa) were employed as upper and lower bounds, respectively (Table S4). The manus and moveable finger reconstructions were exported from Amira as .STL files for analysis. The manus and moveable finger sections for assessed specimens were then imported into 3-matic version 12 (Materialise, Leuven, Belgium) and solid-meshed as distinct solid homogeneous structures consisting of tet-4 elements and the gape angle was set to biologically realistic values between 10–30°. These models were then exported as Nastran files for import into Strand7 (Strand7 Pty Ltd, NSW, Sydney, Australia) FEA software. Material properties used are a Young's modulus of 7 GPa and a Poisson's ratio of 0.3—values used for scorpion cuticle following *Van der Meijden, Kleinteich & Coelho (2012)*. Muscle origins were tessellated as beam elements onto the Nastran models (*Bicknell et al., 2018b*). This was done following successful applications to other arthropods (*Bicknell et al., 2018b*, *2021*) and allowed for a large muscle origin area to be modelled. Muscle forces (Table S3) were assigned to trusses directed toward the insertion site—the most proximal section of the moveable finger. These insertions were treated as static points at the beam terminus. Three denticles along both the manus and moveable finger were constrained in all directions at their most apical node (Fig. 1A); selected denticles were located at the proximal, mid-length, and distal regions of the manus and finger. A hinge between the manus and moveable finger was constructed using two sections: one link on either side of the proximoventral section of the moveable finger. This emulates a simplified action of manus and moveable finger closure. A colour-coded von Mises (VM) stress map was generated after solving models. Loaded Strand7 models are presented as Data S1–S16. These data are found at 10.17605/OSF.IO/GV8J5 Additionally, the analysed .STL files were used to generate 3D PDFs using Tetra4D (Adobe Systems, Mountain View, CA, USA). Scorpion 3D PDFs are available from 10.17605/OSF.IO/GV8J5.

## Fossil models

3D reconstructions of the *Acutiramus bohemicus*, *Erettopterus bilobus*, *Jaekelopterus rhenaniae*, and *Pterygotus anglicus* chelicerae (fixed and free rami) were rendered in Zbrush (Pixologic Inc, Los Angeles, CA, USA). Reconstructions were informed by

examining select fossils and published high resolution images of chelicerae (*Størmer, 1936*; *Waterston, 1964*; *Selden, 1984*; *Chlupáč, 1994*; *Poschmann & Franke, 2006*; *Poschmann & Tetlie, 2006*; *Miller, 2007a*, *2007b*; *Braddy, Poschmann & Tetlie, 2008*; *Lomax, Lamsdell & Ciurca, 2011*; *Kennedy, Miller & Gibling, 2012*; Table S4). These specimens permitted the modelling of chelicerae with anatomically correct dimensions in lateral view. Considering the modern analogues and the 2D preservation of the fossil forms, the degree of cheliceral 'inflation' was informed through examining the three-dimensionality of the modelled scorpion chelae. Furthermore, as there is no evidence to suggest that the fixed ramus is more inflated than the moveable ramus, both pterygotid rami were modelled with a similar degree of inflation. The models were built with internal cavities informed by the scorpion scans and internal cavities of fixed and free rami do not extend into the denticles, following the observations of modern scorpions (*Van der Meijden, Kleinteich & Coelho, 2012*; *Kellersztein et al., 2019*). These reconstructions were required as the examined chelicerae are preserved as compression fossils, with very little relief. There is little to no density difference between the fossil and the host matrix, and scanning these fossils would have produced unreliable 3D data. Furthermore, if the scans were successful, the models would have been retro-deformed, likely using scorpions as the reference. The reconstructions presented here circumvent this limitation of the fossil record. Reconstructions were exported as .STL files from Zbrush. These can be found in Data S17–S20, available from 10.17605/OSF.IO/GV8J5.

The .STL files were scaled to the size of the largest known chelicerae of the respective species to allow for a comparison between presumed adult forms (*Waterston, 1964*; *Chlupáč, 1994*; *Miller, 2007b*; *Braddy, Poschmann & Tetlie, 2008*). This scaling also allowed the calculation of muscle force values for adult pterygotids and permitted the modelling of appendages at 'life size' scales. We did not scale the pterygotid models to the size of scorpion chelae (*Dumont, Grosse & Slater, 2009*; *Bicknell et al., 2021*) as there are orders of magnitude difference in size between scorpion chelae and pterygotid chelicerae. Scaling down chelicerae would likely have introduced allometric errors to the modelling, increasing the uncertainty associated with the pterygotid models. Further, deciding on a particular size to scale the specimens down to is highly subjective. Finally, we were interested in examining the absolute size of chelicerae in this study, thus scaling to the volume of a scorpion chela would have hindered this approach.

After scaling to life size, the .STL files of fixed and free rami were loaded into Meshlab version 2020.12 to estimate the internal volume of the fixed ramus; a proxy for muscle force at the insertion. The internal volume was calculated with the "Ambient Occlusion" filter in Meshlab. All non-occluded elements and the fixed ramus finger were removed to produce the proximal fixed ramus internal morphology. A convex hull of this morphology was then produced and used to calculate the volume and surface area of the internal proximal fixed ramus; a proxy for the size of the adducting muscle. The transverse section of this convex hull representing the muscle mass was used as an estimate of muscle cross-sectional area. These data were $\log_{10}$ transformed and input into the scorpion muscle force regression model to calculate pterygotid muscle force (Table S5).

Force at the muscle insertion for pterygotid species was estimated using data from extant scorpions by two methods: (1) extrapolation of a linear model; and (2) using muscle stress and estimated pterygotid muscle cross-sectional area. We used only the force estimates from the first approach in the FEA models of the pterygotids, as this approach has the fewest assumptions. However, we report and discuss both methods here, as the differences in estimated muscle force obtained may be a cautionary example for other workers using only a single method to estimate muscle force.

In the first approach, all calculated forces at muscle insertions, derived from *in-vivo* pinch force measurements of the scorpions, were used to create a linear model to predict pterygotid muscle force (Tables S6 and S7). The forces at muscle insertions were considered dependent variables against the independent variables of muscle surface and volume. Data were linearized by $log_{10}$ transformation before running the model. The regression coefficients were then used to estimate pterygotid input force at muscle insertions. All statistical regressions were performed in R version 4.1.2 (*R Core Development Team, 2021*; Code S1). Estimated muscle forces employed in the FEMs are shown in Table S3. In the second approach, estimates of muscle stress from *Van der Meijden et al. (2012)* were used to calculate pterygotid muscle force by multiplying these values with pterygotid muscle cross-sectional area.

Boundary, loading, and restraining conditions applied to the cheliceral reconstructions (Fig. 1) were comparable to the scorpion models. Material properties used were the same as scorpions, as the cuticle properties of pterygotid chelicerae are unknown. Muscle origin location, size, and vectors were estimated based on the scorpion comparisons. Muscle origins are located in the fixed ramus proximal region. Similar to scorpion models, muscle origins were tessellated as beam elements. Muscle forces (Table S8) were assigned to trusses directed toward the insertion, in the proximal free ramus section, following *Braddy, Poschmann & Tetlie (2008)*. Similar to scorpion models, three denticles on the fixed and moveable rami were constrained. However, as pterygotid denticles vary in size, the largest denticles were constrained because these points would be the first regions to have contacted prey (Fig. 1B). A colour-coded VM brick stress map was generated after solving models. The loaded Strand7 models are presented in Data S17–S20, available from 10.17605/OSF.IO/GV8J5.

As the pterygotid chelicerae cannot be scanned, there is uncertainty regarding the thickness of the ramus cuticle. Although eurypterid cuticle from the carapace and gnathobases has been sectioned in previous studies (*Dalingwater, 1973, 1975, 1985*; *Bicknell et al., 2018b*), there is no published information on the cuticle thickness of pterygotid chelicerae. Therefore, sensitivity tests for pterygotid models were conducted to explore the impact of cuticle thickness on the biomechanical modelling. *Erettopterus bilobus* fixed and free rami were reconstructed with thinner and thicker cuticle relative to the presented model. Further, this taxon was reconstructed and analysed with cuticle extending into the constrained denticles. These sensitivity test models (loading and boundary conditions outlined below) showed very comparable stress distributions (Fig. S1; Data S17). This suggests that hollowing the rami and denticles has limited impact on the VM stress distribution, but does influence the stress magnitudes.

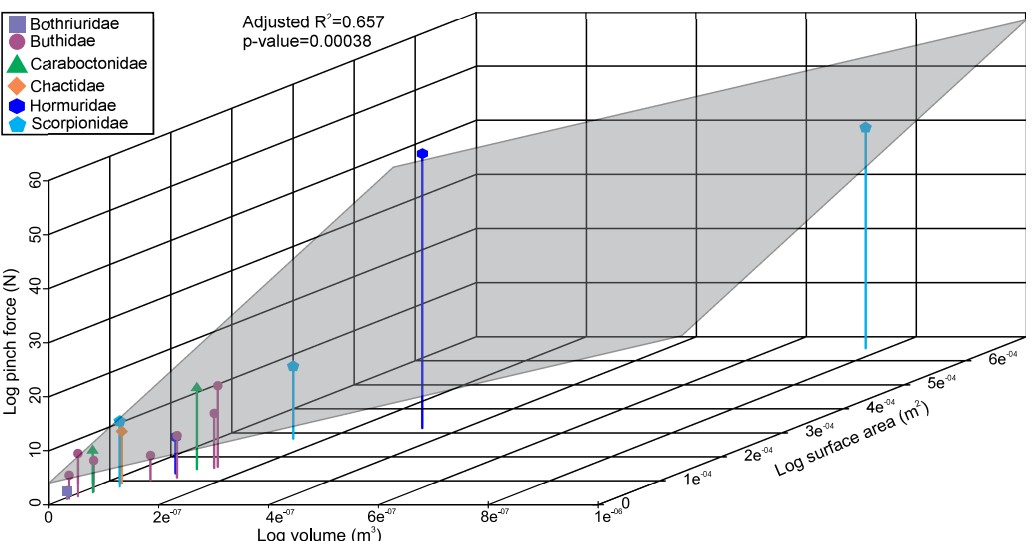

**Figure 2 3D scatterplot of the relationship between scorpion pinch force and muscle volume and surface.** The grey section represents the correlation across the three variables.

Since pterygotid material properties are unknown, it is possible that the values used here may have influenced the VM stresses estimated from the FEMs. Consequently, our results should be considered in comparative contexts only. We therefore limit our interpretation of the models to stress distributions rather than magnitudes, as the former is less sensitive to variation in the assumptions included in our models, such as muscle force, cuticle thickness and material properties.

Mean VM stress values were calculated from all FEMs in Strand7. These values were plotted against the volume of chelae and chelicerae models, exported from Geomagic Studio (3D Systems, Carry, NC, USA). Both values were $\log_{10}$ transformed (Table S9) and plotted in bivariate space.

## RESULTS

The linear model used to estimate muscle force for the scorpions has a statistically significant adjusted $R^2$ value of 0.657 ($p$-value: $3.8e^{-04}$; Fig. 2). Furthermore, estimated muscle volume has a significant correlation to the pinch force estimation (F-value: 29; $p$-value: $1.1e^{-04}$). The pterygotid muscle forces predicted from muscle cross-sectional area and muscle stress were considerably higher than those based on the linear model (Table S5).

The VM stress distributions for the examined scorpion chelae are categorised into three main groups. The first group comprises *Androctonus amoreuxi*, *Caraboctonus keyserlingi*, *Chactas* sp., *Hadogenes paucidens*, *Opistophthalmus boehmi*, and *Scorpio maurus*, each of which exhibit relatively high VM stress across the entire model, with the highest VM stress proximal to the articulation between the manus and moveable finger, and along chelae (Figs. 3A–3F). These forms are typically stouter and have more pronounced proximal manus regions. Notably, *Ca. keyserlingi*, *Ha. paucidens*, and *Sc. maurus* have higher stress

along the moveable finger. The second group consists of *Androctonus australis*, *An. bicolor*, *Grophus flavopiceus*, *Hadrurus arizonensis*, *Hottentotta gentili*, *Leiurus quinquestriatus*, and *Parabuthus transvaalicus*. These models have overall lower VM stress in the manus and variable degrees of VM stress along elongate chelae fingers (Figs. 3G–3M). *Androctonus australis, An. bicolor, Ha. arizonensis, Le. quinquestriatus* and *Pa. traansvalicus* have high VM stress proximal to the articulation between the manus and moveable finger, contrasting with *Gr. flavopiceus* and *Ho. gentili* that have lower VM stress over the entire model. The third group contains *Bothriurus* sp. and *Pandinoides cavimanus*. These models show low VM stress in the proximal manus section, high VM stress at the articulation between the manus and moveable finger, and lower VM stress along stout chelae (Figs. 3N and 3P). Finally, *Opisthacanthus maculatus* is unique, with low VM stress in the proximal manus region and higher VM stress along the length of the chela fingers (Fig. 3O). Overall, these groupings are comparable to those presented by *Van der Meijden, Kleinteich & Coelho (2012)*, where chelae models were standardized for scale using force to surface area ratio.

Distributions of VM stress in the pterygotid models are broadly comparable to those of scorpion FEMs (Fig. 4). Although these appendages are not homologous—representing the first and second limb-bearing segments in eurypterids and scorpions, respectively—the functional analogy of the two can be tested with FEA. The observed VM stress similarity strongly supports the idea that pterygotid chelicerae functioned like scorpion chelae, indicating an informative analogue for these fossil forms. Considering the mean VM stress distributions of the scorpion and pterygotid models, the former have higher mean stress values (Fig. 5). There is limited clustering of family groups in bivariate space, reflecting the range of input force values for the modelled scorpions (Table S9). Taxa within the Buthidae in particular have a large spread of stress within similar volume values. Within the pterygotids, the larger models (*Jaekelopterus rhenaniae* and *Acutiramus bohemicus*) show the lowest mean stress.

Of the pterygotid models, *Erettopterus bilobus* and *Pterygotus anglicus* have comparable VM stress along the fixed and free rami and proximal to the region of articulation between the fixed and free rami (Figs. 4A and 4B). *Pterygotus anglicus* shows high VM stress on the large denticles and *Er. bilobus* has lower VM stress along the distal section of the free ramus. *Jaekelopterus rhenaniae* has the lowest overall VM stress and its only high VM stress region is proximal to the point of articulation between the fixed and free rami and along the proximal section of the free ramus (Fig. 4C). *Acutiramus bohemicus* has high VM stress in the region proximal to the point of articulation between the fixed and free rami and along the proximal section of the free ramus (Fig. 4D). Further, the highest strain areas are located along the oblique, serrated denticles.

In all FEMs, the VM stress is predictably concentrated in areas with high loads, such as muscle insertions and around the fixed vertices on the denticles. In certain scorpion models, higher VM stress areas are visible on the proximal manus region—the location of simulated muscle origins. However, scorpion chelae muscle origins are distributed across the majority of the proximal manus region. As such, VM stress concentration on the proximal manus region reflect where the models were constrained.

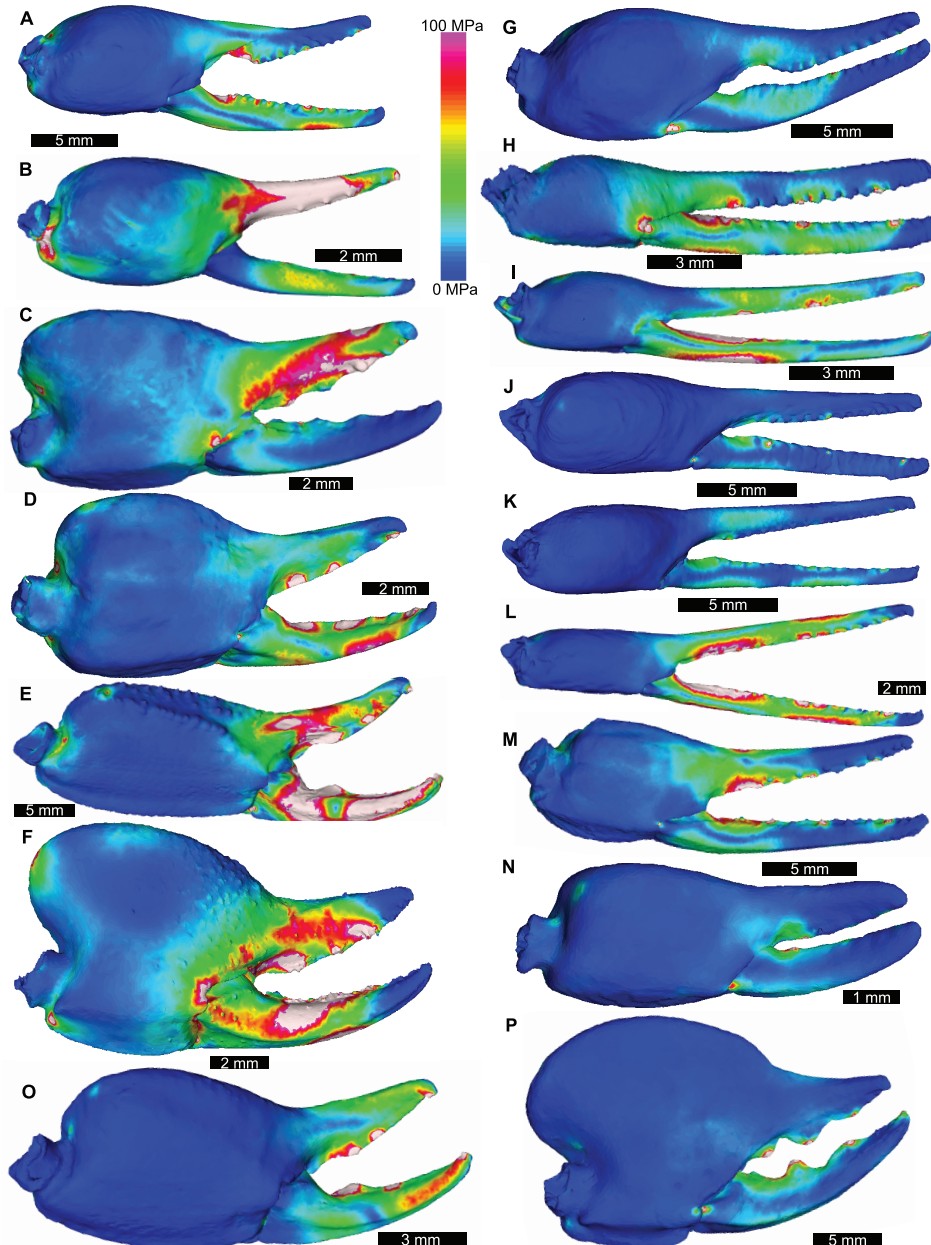

**Figure 3 Lateral views of scorpion finite element models showing von Mises (VM) brick stress maps.**
(A) *Androctonus amoreuxi* Sc450. (B) *Caraboctonus keyserlingi* Sc3009. (C) *Opistophthalmus boehmi* Sc696. (D) *Chactas* sp. Sc999. (E) *Hadogenes paucidens* Sc1041. (F) *Scorpio maurus* Sc1. (G) *Androctonus australis* Sc707 (H) *Parabuthus transvaalicus* Sc 2. (I) *Androctonus bicolor* Sc2623. (J) *Grophus flavopiceus* Sc881. (K) *Hottentotta gentili* Sc172. (L) *Leiurus quinquestriatus* Sc1062. (M) *Hadrurus arizonensis* Sc1004. (N) *Bothriurus* sp. Sc675. (O) *Opisthacanthus maculatus* Sc877. (P) *Pandinoides cavimanus* Sc761. (B, E, F, H, J, K and O) mirrored to align with other appendages. Biomechanical models are found in Data S1–S16.

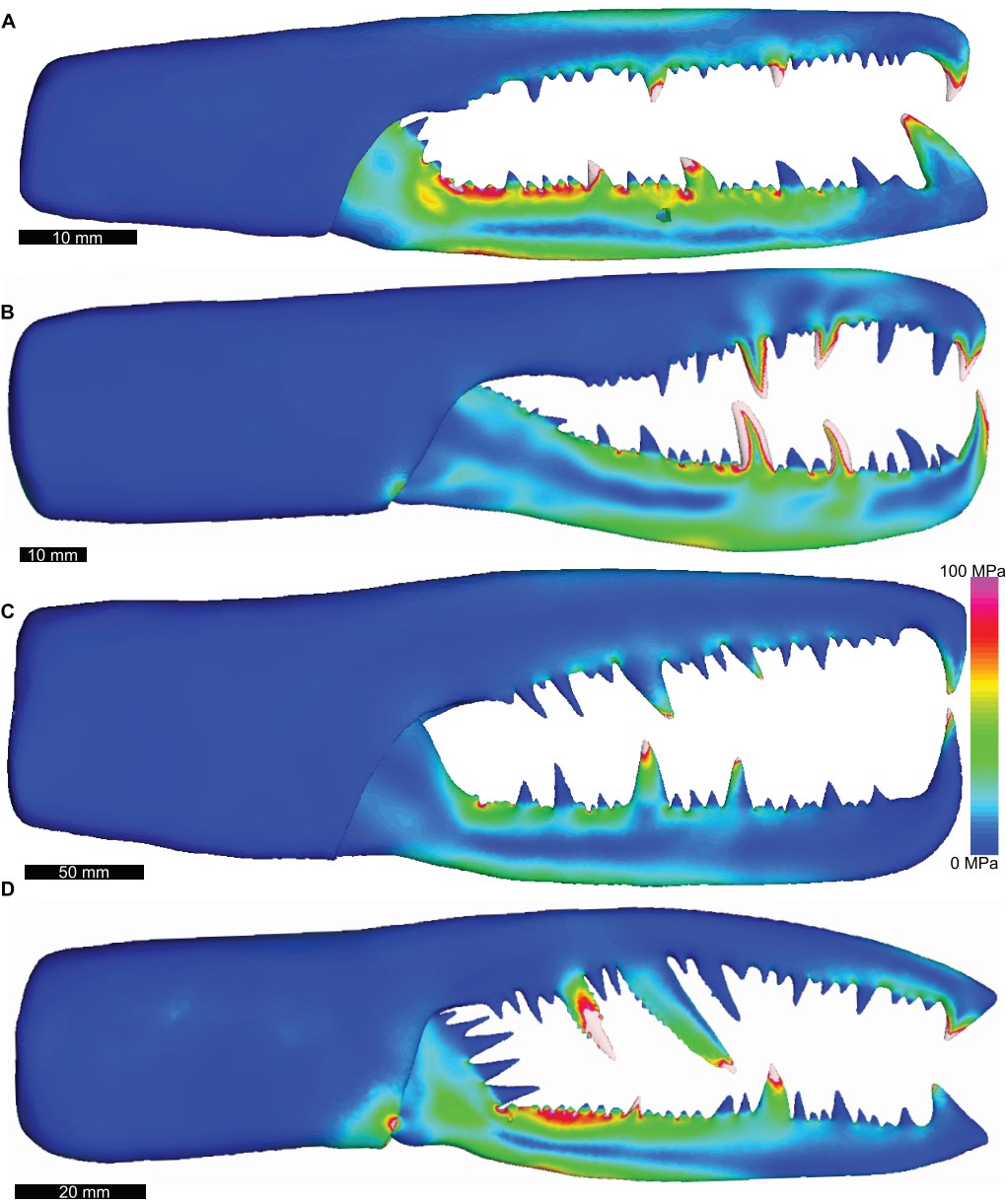

**Figure 4 Lateral views of solved finite element models of assessed pterygotids showing von Mises (VM) brick stress maps.** (A) *Erettopterus bilobus* from the Silurian (latest Llandovery and Wenlock) Patrick and Kip Burn formations, Scotland. (B) *Pterygotus anglicus* from the Devonian (Emsian) Campbellton Formation, Canada. (C) *Jaekelopterus rhenaniae* from the Devonian (Emsian) Klerf and Nellenköpfchen formations, Germany. (D) *Acutiramus bohemicus* from the Silurian (Pridoli) Požáry Formation, Czech Republic. Biomechanical models are found in Data S17–S20.

The large difference in estimated muscle force and stress distributions for the pterygotids between the linear model and the cross-sectional area method is striking. As both methods are commonly used, and both have weaknesses and assumptions, we opted to present the results of both approaches to see if similar values could be predicted. The linear model predicted high muscle force values for the pterygotids. However, as no

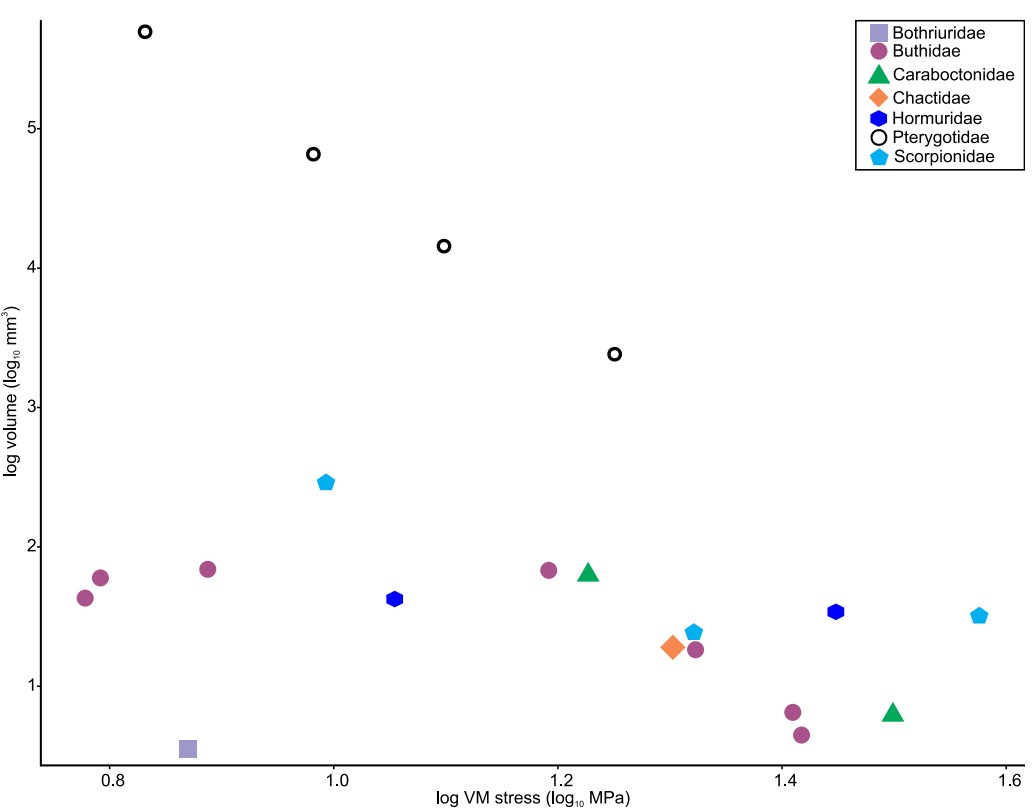

**Figure 5 Scatterplot of the relationship between log transformed VM stress and log transformed model volume for scorpions and pterygotids.**

extant chelicerates have comparably-sized chelae or chelicerae, muscle forces were estimated by extrapolation rather than interpolation. An unavoidable result of extrapolation is increased error. The cross-sectional area method of muscle force estimation has more explicit assumptions, each with a level of uncertainty. The range of lowest to highest muscle stress estimates illustrate this uncertainty (Table S5). For these reasons, we did not use the muscle force estimates derived from cross-sectional area in the FEMs, and did not focus on absolute magnitudes of stress or force in our analysis and discussion.

## DISCUSSION

The results of the scorpion FEAs show similar groupings as in *Van der Meijden, Kleinteich & Coelho (2012)*. The differences in magnitudes of VM stresses (Fig. 3) indicate that some species (*e.g.*, *Pandinoides cavimanus*, *Ophistacanthus maculatus*, *Hottentotta gentili*) could exhibit a higher 'safety factor' (*i.e.*, a measure of how much stronger a morphology needs to be compared to the input forces; *Hicks & Wang, 2021*) than others (*e.g.*, *Caraboctonus keyserlingi*, *Hadogenes paucidens*, *Parabuthus transvaalicus*), although the pattern does not seem to correspond to defensive behaviour (*Van der Meijden et al., 2013*) or relative pinch force (*Simone & Van der Meijden, 2018*). Considering the modelled scorpions in the context of possible ecomorphs, there is limited overlap between groups identified here and recently proposed ecomorphologies (*Coelho et al., 2022*). This suggests that the FEMs

produced here present more of an insight into the effectiveness of these feeding tools, rather than inform on possible ecological groupings and associated microhabitats. Combining these biomechanical analyses with detailed examination of scorpion life modes will undoubtedly uncover new links between chelae mechanics and the diets of these ecomorphs. However, this is beyond the scope of this work, as we have focused on using scorpions to inform aspects of the pterygotid models. On this point, it is worth considering other possible modern analogues for understanding and modelling pterygotids. Of particular note are the camel spiders (Solifugae), which have large, dentate chelicerae that can grasp and disarticulate prey for consumption (*Van der Meijden et al., 2012*). These chelicerae are anteriorly directed, but do not extend notably beyond the prosoma (*Harms & Duperre, 2018*). While they may be informative for understanding how chelicerae can break prey, they are less useful than scorpion pedipalps as functional analogues for pterygotid chelicerae.

The morphology of pterygotid chelicerae suggests that these structures were used in capturing prey for subsequent mastication by the coxal gnathobases (*Waterston, 1964*; *Miller, 2007a*). The solifuge comparisons made above suggest that the dentition of pterygotid chelicerae may have allowed for some initial tearing of prey. This inferred feeding mode aligns with the co-occurrence of pterygotids and proposed prey species, including other eurypterids, fishes, and a likely array of soft-bodied animals that are not preserved (*Rolfe, 1973*; *Chlupáč et al., 1980*; *Kennedy, Miller & Gibling, 2012*). Furthermore, links between cheliceral and lateral compound eye morphologies have been drawn to propose an apex predatory life mode for some pterygotids (*Anderson et al., 2014*; *McCoy et al., 2015*).

Results of the FEA support previous inferences that pterygotid chelicerae, while having a similar overall morphology, had differing functional capabilities (*McCoy et al., 2015*). The comparable VM stress distributions between the chelicerae of *Erettopterus bilobus* and *Pterygotus anglicus* suggest that these taxa would have experienced similar stresses while feeding and therefore probably targeted similar prey. Based on the known faunal assemblages that co-occur with these pterygotid species (*Dunlop, Braddy & Tetlie, 2002*; *Miller, 2007b*; *Lebedev et al., 2009*; *McCoy et al., 2015*; *Fyffe, Johnson & van Staal, 2016*; *Blieck, 2017*; Tables S10 and S11), coupled with the large body lengths of *Er. bilobus* (0.7 m) and *Pt. anglicus* (1.6 m), proposed prey includes other eurypterids and fishes; in the case of the Devonian *Pt. anglicus*, the latter could include armoured forms, such as ostracoderms and placoderms. This is further supported by the moderate to high visual acuity in these pterygotid genera (*McCoy et al., 2015*), allowing them to identify and pursue such mobile, possibly smaller prey. Indirect evidence for *Er. bilobus* consuming fish includes coprolites rich in agnathan fragments from the Lesmahagow Inlier that have been previously attributed to eurypterid predation (*Rolfe, 1973*; *Selden, 1984*). It therefore seems that *Er. bilobus* and *Pt. anglicus* were apex predators within their respective ecosystems (*Selden, 1984*; *Dunlop, Braddy & Tetlie, 2002*; *Kennedy, Miller & Gibling, 2012*). This contradicts a previous suggestion that *Er. bilobus* was a more generalised predator than *Pt. anglicus* (*McCoy et al., 2015*).

The FEM of *Acutiramus bohemicus* produced a unique VM stress distribution associated with its different cheliceral morphology. All *Acutiramus* species have elongate (some hypertrophied), oblique, anteriorly-directed, serrated denticles (*Chlupáč, 1994*; *Laub, Tollerton & Berkof, 2010*; *McCoy et al., 2015*). High VM stress along the free ramus and within the constrained denticles suggest that *Acutiramus* was not well adapted to capturing armoured or thick-shelled prey and may have experienced failure when doing so (*Laub, Tollerton & Berkof, 2010*). Indeed, *Acutiramus* does not co-occur with a diverse fish fauna, but rather an array of other eurypterid taxa (*Laub, Tollerton & Berkof, 2010*; *Anderson et al., 2014*; Table S12). The angled, serrated denticles would be more consistent with piercing eurypterid cuticle (*Laub, Tollerton & Berkof, 2010*; *Anderson et al., 2014*; *Fyffe, Johnson & van Staal, 2016*) and the proximal denticles on the free ramus may have impaled prey (*Laub, Tollerton & Berkof, 2010*). After impaling, the prey would have been sliced by the serrated denticles as the chelicerae closed. The serrations documented by *Laub, Tollerton & Berkof (2010)* are proximally orientated, suggesting that after impaling, a victim would have had to tear itself from the serrations, likely causing more damage. This conforms to the idea that *Acutiramus* was likely an ambush predator or scavenger that fed on soft-bodied and lightly cuticularized taxa, based on its limited vision and cheliceral morphology (*Anderson et al., 2014*; *McCoy et al., 2015*).

The *Jaekelopterus rhenaniae* model has overall low VM stress compared to the other pterygotid models. These data, combined with a proposed 2.5 m body length (*Braddy, Poschmann & Tetlie, 2008*), high visual acuity (*McCoy et al., 2015*; *Poschmann, Schoenemann & McCoy, 2016*), good swimming abilities (*Plotnick & Baumiller, 1988*; *Tetlie, 2007*), and a diverse co-occurring fauna containing several eurypterid and armoured fish species (*Fyffe, Johnson & van Staal, 2016*; *Poschmann, Schoenemann & McCoy, 2016*; Table S13), suggest that *Ja. rhenaniae* was capable of capturing large, highly mobile, armoured prey. The robust chelicerae would have enabled the initial grabbing and manipulation of food items, while the heavily sclerotised gnathobases on large coxae would have been employed in prey mastication (including crushing of thick cuticle or biomineralised structures), comparable to extant xiphosurids (*Waterston, 1964*; *Botton, 1984*; *Poschmann, Bergmann & Kühl, 2017*; *Bicknell et al., 2018a*, *2018b*). The large reinforced chelicerae, giant body size, and phylogenetically-derived position of *Ja. rhenaniae* (Fig. 6; *Braddy, Poschmann & Tetlie, 2008*) appear to represent a peak in pterygotid cheliceral evolution that coincides with the rapid diversification of placoderms (*Randle & Sansom, 2019*). As such, predatory Devonian eurypterids, epitomised by *Ja. rhenaniae*, likely developed a toolkit to target armoured fish in the form of robust chelicerae for grabbing prey and stout gnathobases for masticating said prey (*Poschmann, Bergmann & Kühl, 2017*), as opposed to eurypterids driving the radiation of placoderms (*contra Romer, 1933*). The extinction of pterygotids in the Late Devonian is consistent with them ultimately being outcompeted by vertebrates (particularly jawed fishes), cephalopods, and various other groups during the Devonian Nekton Revolution (*Lamsdell & Braddy, 2010*; *Klug et al., 2010*, *2018*), as well as falling victim to environmental changes (*Lamsdell & Selden, 2017*).

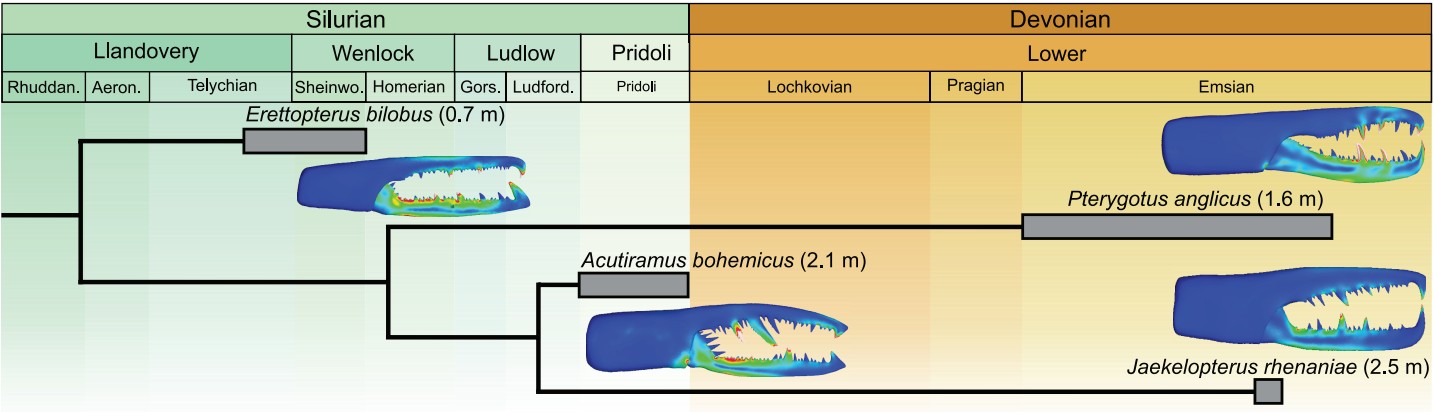

**Figure 6 Simplified phylogeny of analysed pterygotid taxa showing total body lengths (in metres), associated time ranges and FEMs.** Derived from *Braddy, Poschmann & Tetlie (2008*; fig. 2) and *Lamsdell & Selden (2017*; fig. 1).

# CONCLUSIONS

Finite element models of chelicerae from four pterygotid eurypterids (*Acutiramus bohemicus*, *Erettopterus bilobus*, *Jaekelopterus rhenaniae*, and *Pterygotus anglicus*) are presented and compared with FEMs of modern scorpion chelae. Overall similarity in VM stress distributions indicates that pterygotid chelicerae are functionally analogous to scorpion chelae, as suggested by the morphological similarity of these structures. Considering the pterygotid VM stress distributions in the context of their palaeoecology (including visual capabilities) and overall cheliceral morphology, we have demonstrated, with FEA, the morpho-functional ability of these ancient marine predators. We conclude that *Er. bilobus* and *Pt. anglicus* were apex predators within their respective ecosystems and likely targeted other eurypterids and fishes. Stress distributions of the *Ac. bohemicus* model, together with its unique denticle morphology, suggest that the chelicera of this species was adapted to piercing and slicing the cuticle of other eurypterids. Finally, *Ja. rhenaniae*, the largest known pterygotid, experienced low VM stress across the chelicera, suggesting that it was well adapted to capturing large, highly mobile, armoured prey. These results demonstrate how 3D FEA can be applied across a range of morphologies and used to explore the mechanical performance of extinct predatory arthropods.

# ACKNOWLEDGEMENTS

We thank Katrina Kenny for producing the 3D reconstructions of the fossil taxa, Thomas Kleinteich for his help in scanning the scorpion chelae, and Jason Dunlop and Carolin Haug for their constructive reviews.

## Funding

This research was supported by funding from an Australian Research Council Discovery Project grant (DP200102005 to John R Paterson, Stephen Wroe, and Gregory D. Edgecombe), and a UNE Postdoctoral Research Fellowship (to Russell D. C. Bicknell). Arie

van der Meijden is financed through FCT, I.P. under contract number DL57/2016/CP1440/CT0009. Yuri Simone was funded by a PhD scholarship from Fundação para Ciênciae Tecnologia (SFRH/BD/136934/2018). The funders had no role in study design, data collection and analysis, decision to publish, or preparation of the manuscript.

## Grant Disclosures

The following grant information was disclosed by the authors:
Australian Research Council Discovery Project: DP200102005.
UNE Postdoctoral Research Fellowship.
FCT, I.P: DL57/2016/CP1440/CT0009.
Fundação para Ciênciae Tecnologia: SFRH/BD/136934/2018.

## Competing Interests

The authors declare that they have no competing interests.

## Author Contributions

- Russell D. C. Bicknell conceived and designed the experiments, performed the experiments, analyzed the data, prepared figures and/or tables, authored or reviewed drafts of the article, and approved the final draft.
- Yuri Simone conceived and designed the experiments, performed the experiments, analyzed the data, prepared figures and/or tables, authored or reviewed drafts of the article, and approved the final draft.
- Arie van der Meijden conceived and designed the experiments, performed the experiments, analyzed the data, authored or reviewed drafts of the article, and approved the final draft.
- Stephen Wroe conceived and designed the experiments, performed the experiments, authored or reviewed drafts of the article, and approved the final draft.
- Gregory D. Edgecombe conceived and designed the experiments, authored or reviewed drafts of the article, and approved the final draft.
- John R. Paterson conceived and designed the experiments, authored or reviewed drafts of the article, and approved the final draft.

## Data Availability

The data is available at OSF: Bicknell, Russell D. 2022. "Supplemental Data for Biomechanical Analyses of Pterygotid Sea Scorpion Chelicerae Uncover Predatory Specialisation within Eurypterids." OSF. October 28. DOI 10.17605/OSF.IO/GV8J5.

## Supplemental Information

Supplemental information for this article can be found online at http://dx.doi.org/10.7717/peerj.14515#supplemental-information.

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
