# Peer review of "Biomechanical analyses of pterygotid sea scorpion chelicerae uncover predatory specialisation within eurypterids"

_PeerJ, doi:10.7717/peerj.14515_

## Round 0.1 · original submission · Minor Revisions

Both reviewers had a favorable view of the paper but have identified a number of minor issues that I do not think should be too difficult to address and I think it would be worthwhile to make the changes suggested. When the authors resubmit their revised manuscript, please provide a tracked changes version of the manuscript, as well as a letter detailing how they have addressed the various comments and suggestions of the reviewers.

·

Basic reporting

This manuscript models the biomechanics of the cheliceral claws of pterygotid eurypterids. As the authors correctly state, these are one of the largest groups of predatory arthropods ever to have lived. The authors use the pedipalp claws of scorpions as the closest modern analogy and apply finite element mapping to the fossils in order to infer how the cheliceral claws of four different pterygoid eurypterid genera may have functioned.

I cannot comment in detail on the methods of the biomechanical analysis (I hope this will be checked by other reviewers), but the work appears to have been carried out compentently by authors who have previous published experience in this field with both fossil and living arthropods. The manuscript is well-written throughout, properly structured and the literature review is good. The figures are nice and the 3D pdfs opened correctly from the download link and could be manually rotated as expected.

While there is always going to be an element of uncertainty in reconstructing the biomechanics and mode of life of extinct animals, and several assumptions are necessary, I think the manuscript does offer a useful investigation showing how the structure of living arachnids can be used to help understand possible feeding ecologies in fossil taxa. It also introduces the hypothesis that different genera of pterygotid eurypterids may have had different feeding stratergies, reflected in different patterns of cheliceral morphology.

Overall I think the manuscript merits publication subject to mostly minor revision (see below).

Experimental design

I understand why the authors are using scorpions as their main proxy, as the scorpion pedipalpal claws are similar in size and shape to the pterygotid chelicerae. I think this approach is OK and will yield useful data on, e.g., the likely strength of the "pinch". However, some of the authors (Van der Meijden 2012b) have looked at camel spiders (Solifugae) and I would be tempted to integrate comparisons from this group of arachnids, at least in the discussion of the pterygotids' functional morphology (e.g. lines 321-322).

If you have a look at the chelicerae of camel spiders

e.g. <https://phys.org/news/2015-06-formidable-camel-spider-jaws.html>

you can see that they often have strong dentition which is really quite similar to that of the more derived pterygotids, with spines/teeth angled vertically or slightly anteriorly on both the fixed and free ramus. As far as I'm aware, the camel spiders use their chelicerae to tear prey apart by holding the captured animals in these teeth and moving the chelicerae backwards and forwards in opposition against each other to essentially rip the prey apart. Maybe there are some functional morphological studies on camel spiders in the literature?

I do wonder if pterygotids used their chelicerae in a similar way, i.e. not only for initially capturing/restraining prey like the scorpion pedipalps, but also for actively tearing it into smaller pieces prior to final mastication in the gnathobases. In this sense the scorpion pedipalp claw as a proxy is slightly weakened by the fact that it either lacks strong dentition (or only has a few crenulated ridges) and is used primarily to grab prey and bring it close to the mouth where it is then masticated by the chelicerae.

As I say, I think the scorpion model is approrpiate for estimating grasping strength, but could be refined in terms of functionality by considering other modern arachnids.

Validity of the findings

I think the overall validity of the findings are sound, and as noted above the authors offer a plausible explanation for why the four genera have different cheliceral morphologies.

I would, however, pick up a point (e.g. lines 323-325) about the inferred diet. Palaeoecology is always difficult to infer as we will never know the full fauna and tropic relationships between the taxa. You suggest that pterygoids were feeding on other eurypterids and/or placoderms, but is there evidence for, say, molluscs or conodonts (or anything else) in the pterygotid-bearing faunas? I could envisage pterygotids preying on for example the eel-like conodont animals too, as they would be fairly soft prey items, reducing the risk of damage to the cheliceral teeth. It would be a safer choice than a placoderm! It might be similar to scorpions or camel spiders eating things like insect larvae today.

I would thus be cautious about restricting discussions of potential prey to only a couple of taxa. Many arachnids are fairly generalist predators and perhaps eurypterids were not too fussy about what they caught either.

Additional comments

Minor corrections

INTRODUCTION

line 46: maybe delete "particularly humans..." as not all of the following references refer to primates. Sentence seemed a bit misleading.

line 49 "(2018a, 2021, 2021)" delete second 2021 ?


RESULTS

line 275: "Within the eurypterids..." [not eurypterid]

line 293: "...reflect where..." [not were]


DISCUSSION

line 318: I'm not sure "habitats" is the correct term here as it refers to where the animals live. I think you mean "mode of life" or "mode of feeding" for how the claw morphology relates to prey capture.


CONCLUSION

line 382: should the subtitle be Conclusion or Conclusions ?


REFERENCES

line 472: Esteve et al. 2021 doesn't appear to be cited in the text.

line 460: Should there be a place of publication (city?) for the Snodgrass textbook. I would have expected it to be something like "Cornell University Press, CITY?"


FIGURE LEGENDS

line 610: maybe define FEM here again as "finite element models" as this is the first time it is mentioned in the figures?

line 611: you have one "SC", but the others are "Sc"

line 611: in a wider context I was not sure what these Sc numbers refer to. Specimen numbers from a collection? A biomechanical parameter? Its not clear from the figure legend what these are.

·

Basic reporting

The authors are aware of the limitations of their method and clearly explain what they are doing and in how far conclusions are possible, which is very important. In general, everything is well written and explained, but there are some important infos that still need to be added (see below, Additional comments). I think that the study can be published after minor to moderate revision.

Experimental design

no comment

Validity of the findings

no comment

Additional comments

Detailed comments:

18: Not the genera have characters, but their representatives; better rephrase to “Representatives of this family all have...”

24: Is it taxa or species? Be more specific if possible (also in l. 79)

33: no “the” in front of group names (here Pterygotidae, but also check the remaining text) for philosophical reasons, or you write “the group”

49: Are there any other references for 3D FEA on arthropods besides Bicknell? And are there any studies on 3D FEA on other groups besides vertebrates and arthropods, maybe mollusks?

67: Maybe add some other references that scorpions and eurypterids are closer related, though it seems a bit trivial, as it is clear that both are chelicerates, but if mentioned, it needs to be properly referenced.

101: Do the specimens have repository numbers? If yes, add them, but I am aware that in extant collections not all specimens get numbers.

123: “and solid-meshed as distinct solid homogeneous structures in tet-4 elements”; the sentence is a bit unclear to me, maybe rephrase; also the following sentences are partly cryptic for someone not familiar with the method, so the authors may want to consider to rephrase some.

123: “Here the gape angle to was set to”; I assume there is a superfluous “to”

147: Which selected fossils have been studied? From which collections and which repository numbers? Are normal images of these specimens available in the supplements? If not, these images and information on the specimens need to be added for transparency, as these are the raw data on which the analysis is based.

207: Are there any options to get information on the original material composition in any sea scorpion? There are sea scorpions with organic preservation, maybe these provide data on the material, e.g. if heavy metals are included in some areas as is known from some extant arthropods. Of course, it would be necessary to look at a carcass and not a moult, as in the latter the material composition changed prior to moulting.

270: “indicating and informative analogue”; should be “an” instead of “and” I assume

275: eurypterids (plural)

293: “were the models”; should be “where”

354: Further above, the authors explain that the cuticle thickness of eurypterids is not known, so this is contradictory here. Also, when looking at extant marine arthropods, e.g. Limulus or also decapod crustaceans, the cuticle gets thicker during ontogeny and also incorporates hardening substances. Hence, the sentence needs to be rephrased (also check in the conclusions).

379: The term “Devonian Nekton Revolution” should be mentioned here, together with suitable references (especially Klug et al. 2010 Lethaia).

Fig. 1:
I would write “model” instead of “models” (2x). “Key” is not necessary, delete the word. The arrows on the constrained denticles are very small, maybe make the lines thinner so that they are still recognisable as arrows.

Carolin Haug, LMU Munich

---

## Round 0.2 · accepted · Accept

The authors have done a very good job addressing the comments from reviewers and in my estimation, the paper is now ready to be accepted and move to the next stage for publication.